# FOURIER ORDINARY DIFFERENTIAL EQUATIONS

## ABSTRACT

Continuous models such as Neural Ordinary Differential Equations (NODEs) are powerful approaches for modeling time series data, known for their ability to capture underlying dynamics and generalization. Current continuous models focus on learning mappings within finite-dimensional time domains, raising two critical questions for enhancing their effectiveness. First, Is the time domain the optimal representation for capturing the underlying patterns and features in time series data? Second, how can we maintain granularity while benefiting from the generalization capabilities of continuous models? To address the first question, we propose a novel approach for learning dynamics in the Fourier domain. In contrast to the time domain, each point in Fourier space summarizes the original signal at a specific frequency, enabling more comprehensive data representations. Additionally, time differentiation in the Fourier domain simplifies the modeling of dynamics as it becomes a multiplication operation. To answer the second question, we introduce element-wise filtering, a method designed to compensate for the bias of continuous models when fitting discrete data points. These techniques culminate in the introduction of a new approach—Fourier Ordinary Differential Equations (FODEs). Our experiments provide compelling evidence of FODEs' superiority in terms of accuracy, efficiency, and generalization capabilities when compared to existing methods across various time series datasets. By offering a novel method for modeling time series data capable of capturing both short-term and long-term patterns, FODEs have the potential to significantly enhance the modeling and prediction of complex dynamic systems.

## 1 INTRODUCTION

Neural Ordinary Differential Equations (Chen et al., 2018) redefine the dynamics of hidden state representations by utilizing ordinary differential equations (ODEs). In contrast to conventional architectures with discrete hidden layers, NODEs exhibit a continuous depth that enables them to adapt to varying inputs while trading numerical precision for computational efficiency. Within NODEs, the dynamics of hidden features $h(t) \in \mathbb{R}^N$ are governed by an ODE parametrized by a neural network $f(h(t), t, \theta) \in \mathbb{R}^N$ with learnable parameters $\theta$ (Xia et al., 2021; Kidger, 2022). This ODE captures the temporal evolution of a quantity of interest, represented by the vector $h$, and describes how it changes over time (Biloš et al., 2021; Dupont et al., 2019; Guo et al., 2023). By specifying an initial value $h(t_0)$, we can determine the state of the dynamic system at any desired time $t_1$.

$$h(t_1) = h(t_0) + \int_0^{t_1} f(h(t), t, \theta)dt = ODESolver(h(t_0), f, t_0, t_1, \theta). \tag{1}$$

When modeling time series data, it is crucial to consider certain factors. Firstly, capturing global patterns is essential, as time series data often exhibit long-term trends or cyclical patterns that span the entire dataset. Effectively capturing these global patterns is vital for accurate modeling. Secondly, identifying long-term relationships among variables is challenging, as simple models may struggle to capture such complex dependencies.

In the context of Neural ODEs, which learn mappings in the time domain, two key questions arise. First, is the time domain the optimal representation for capturing underlying patterns and features in time series data? Second, how can we reconcile the discrete nature of time series data with the generalization capabilities of continuous models?

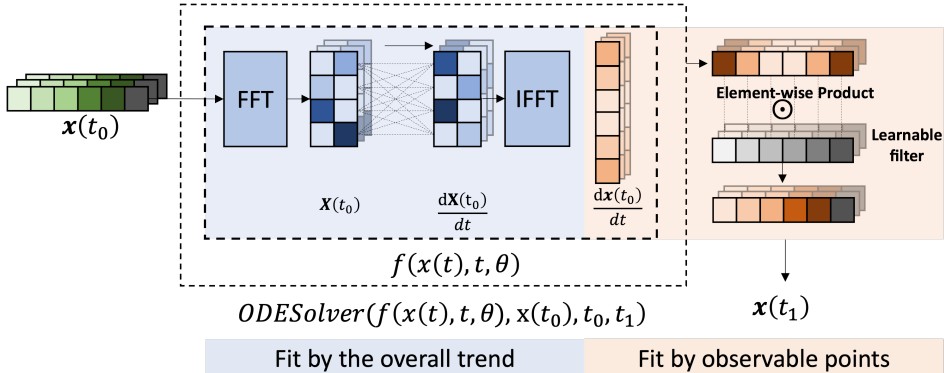

Figure 1: An overview of our predictive model. Input is the time series data $x$, with an implicit initial time $t_0$. (Blue part) The FFT and IFFT are used to convert the data from time-space to Fourier space and from Fourier space to time space, respectively. The dynamics are learned in the Fourier space. (Orange part) The element-wised filter is applied to the output from the ODESolver to maintain the granularity of the original data.

To address the first question, exploring the Fourier domain emerges as a potential solution. By analyzing the frequency components of data using the Fourier transform, we can capture global patterns and identify long-term relationships between variables (Rao et al., 2021; Zhou et al., 2022). The Fourier domain represents time series data as frequency components, where each component represents a specific frequency present in the data. Analyzing these components allows us to identify dominant frequencies and patterns that may not be evident in the time domain. For example, the Fourier transform reveals the frequency and amplitude of a periodic signal, such as a sine wave.

Furthermore, the frequency domain facilitates the analysis of relationships between different frequencies, unveiling long-term dependencies between variables that elude simple time-domain models. Recent studies (Lee-Thorp et al., 2021; Zhou et al., 2022) demonstrate the efficacy of incorporating Fourier analysis into models, achieving competitive performance on benchmark tasks such as the GLUE benchmark (Wang et al., 2018) and capturing global properties of time series using Transformer-based architectures (Vaswani et al., 2017).

The second question revolves around preserving the granularity of the original data when fitting continuous models. Adjustments are necessary to align the model predictions with the discrete data points. One approach is element-wise filtering, accomplished through element-wise multiplication, also known as the Hadamard product (Horn, 1990). Element-wise filtering plays a vital role in refining model outputs by introducing correction factors learned from the data or prior knowledge (Xing et al., 2018; Rao et al., 2021; Chen et al., 2019; Dzanic & Witherden, 2022). Moreover, it can be leveraged to emphasize specific features or components in the prediction by multiplying the model's output with a weight vector or attention mask learned from the data or prior knowledge. This approach enhances the model's focus on relevant features or regions of interest.

To address these considerations, we propose Fourier Ordinary Differential Equations (FODEs) that combine the strengths of both discrete and continuous models (see Figure 1). FODEs leverage the Fourier domain to capture global patterns and long-term relationships while preserving the granularity of the original data through element-wise filtering. This novel approach enhances the effectiveness of time series modeling and holds promise for various applications in neural signal processing, time series forecasting, and other related domains.

## 2 METHOD

### 2.1 DISCRETE FOURIER TRANSFORM

We commence by introducing the Discrete Fourier Transform (DFT) (Winograd, 1978; Wang, 1984), a fundamental tool in digital signal processing. The DFT is applied to a sequence of $N$ complex

numbers $x[n]$, where $0 \leq n \leq N - 1$, to convert it into the frequency domain. The 1D DFT is defined as follows:

$$X[k] = \sum_{n=0}^{N-1} x[n]e^{-i2\pi kn/N}, \tag{2}$$

where $i$ denotes the imaginary unit. The DFT maps the input sequence $x[n]$ to its spectrum $X[k]$ at the frequency $\omega_k = \frac{2\pi k}{N}$. Since $X[k]$ repeats on intervals of length $N$, it suffices to consider the values of $X[k]$ at $N$ consecutive points $k = 0, 1, \cdots, N - 1$.

The DFT is a one-to-one transformation, meaning that given $X[k]$, we can recover the original signal $x[n]$ using the inverse DFT (IDFT):

$$x[n] = \frac{1}{N} \sum_{k=0}^{N-1} X[k]e^{i2\pi kn/N}. \tag{3}$$

The DFT's significance lies in its application to signal processing algorithms, particularly within two important contexts. Firstly, the DFT operates on discrete inputs and produces discrete outputs, making it computationally suitable for digital signal processing. Secondly, the development of efficient algorithms, such as the Fast Fourier Transform (FFT) (Brigham, 1988), has revolutionized DFT computation. The FFT exploits the symmetry properties of the DFT and employs a divide-and-conquer approach (Gorlatch, 1998), recursively breaking down the DFT into smaller subproblems. This approach drastically reduces the computational complexity from $O(N^2)$ to $O(N \log N)$ (Rao et al., 2021). Notably, the inverse DFT, which exhibits a similar structure as the DFT, can also be efficiently computed using the inverse Fast Fourier Transform (IFFT). These advancements in DFT and FFT techniques have significantly enhanced the efficiency and practicality of signal-processing algorithms across various domains.

## 2.2 CONSTRUCT DYNAMICS IN FOURIER SPACE

By delving into the frequency components of data in the Fourier domain, we can capture comprehensive patterns that extend across the entire dataset and uncover lasting connections between variables (Rao et al., 2021; Zhou et al., 2022). In the Fourier domain, time series data is represented as a sequence of frequency components, each representing a unique frequency present in the data. Analyzing these frequency components enables the identification of dominant frequencies and patterns that may not be easily discernible in the time domain. Moreover, the frequency domain facilitates the exploration of relationships between different frequencies, unveiling long-term associations between variables that elude simplistic time-domain models.

To leverage these insights, we introduce Fourier Ordinary Differential Equations (FODE) that learn dynamics in the Fourier domain. Given the input data $x$ and an implicit initial time $t_0$, its corresponding representation $X$ in the Fourier domain can be represented by:

$$X(k, t_0) = \sum_{n=0}^{N-1} x(n, t_0)e^{-i2\pi kn/N}. \tag{4}$$

$X$ is the complex tensor and represents the spectrum of $x$. For real input $x[n]$, its DFT is conjugate symmetric (Rao et al., 2021), i.e. $X[N - k] = X^*[k]$. The reverse is true as well: if we perform IDFT to $X[k]$ which is conjugate symmetric, a real discrete signal can be recovered. This property implies that the half of the DFT $\{X[k] : 0 \leq k \leq \lceil \frac{N}{2} \rceil\}$ contains the full information about the frequency characteristics of $x[n]$. Suppose the real and imaginary parts of $X[k]$ are $X[k]^{real}$ and $X[k]^{imag}$, respectively. The complex numbers represent the spectrum of the signal in the Fourier domain, which provides information about both the amplitude and phase of the frequency components present in the original signal. The magnitude of the complex numbers represents the strength or magnitude of each frequency component, while the phase represents the phase shift or timing information associated with each component. We concatenate the $X[k]^{real}$ and $X[k]^{imag}$ together by

$$X^{info} = X[k]^{real} \oplus X[k]^{imag}, \tag{5}$$

where the $\oplus$ represents the concatenate symbol. Thus, $X^{info}$ contains the real and imaginary part information without the imaginary symbol $i$. We aim to learn a mapping rule $g : X \to Z$:

$$Z^{info} = g(X^{info}). \tag{6}$$

We do this by means of a basic neural network, i.e., $g(\cdot)$ is a basic neural network, which can be implemented by a Multilayer Perceptron (MLP) (Haykin, 1998) in practice.

The obtained tensor $Z^{info}$ contains the information in the Fourier domain, thus we separate it to extract the real and "imaginary" part by:

$$Z[k]^{real} \oplus Z[k]^{imag} = Z^{info}. \tag{7}$$

Note that the $Z[k]^{imag}$ does not contain the imaginary unit $i$, so we construct the complex tensor by applying an imaginary unit $i$ on the "imaginary" part $Z[k]^{imag}$ and obtain a real complex tensor $Z[k] = Z[k]^{real} + iZ[k]^{imag}$. The complex tensor $Z[k]$ can be seen as a representation in the Fourier space. Finally, we can map back to the time domain by applying the inverse Fast Fourier Transform (IFFT):

$$z[n] = \frac{1}{N} \sum_{k=0}^{N-1} Z[k] e^{-i2\pi kn/N}. \tag{8}$$

## 2.3 FOURIER ORDINARY DIFFERENTIAL EQUATIONS

In the last section, we construct dynamics in Fourier space. Now we can combine them together and obtain the dynamic function $f$:

$$f(x,t) = f\left(\frac{1}{N} \sum_{k=0}^{N-1} g(X(k,t)) e^{-i2\pi kn/N}\right). \tag{9}$$

By leveraging the Fast Fourier Transform (FFT), a basic neural network $g(\cdot)$, and the Inverse Fast Fourier Transform (IFFT), we construct the dynamic function $f(x,t)$. This function represents the change in dynamics in the Fourier space and is a function of the data $x$ and an artificially introduced time $t$, rather than an explicit time present in the data.

To obtain the final state of the system at a given time $t_1$, we solve an initial value problem (IVP) using an ODE solver. The IVP is formulated as:

$$x(t_1) = x(t_0) + \int_0^{t_1} f(x,t)dt = ODESolver(x(t_0), f, t_0, t_1, \theta), \tag{10}$$

where $x(t_0)$ represents the initial state, $f$ is the dynamic function, and $\theta$ represents the parameters in the neural network $g(\cdot)$. The ODE solver approximates the solution of the ODE by iteratively integrating it forward in time, starting from the initial condition $x(t_0)$.

One advantage of ODE-based models is their inherent invertibility, allowing us to reverse the integration limits or integrate the negative of the function $f$. This property enables the computation of gradients for the solutions of initial value problems with respect to both the parameters $\theta_f$ and the initial values $x(t_0)$. The *Adjoint Sensitivity Method* (Pontryagin et al., 1961), based on reverse-time integration of an extended ODE, is employed to calculate these gradients. The ODE solver plays a crucial role in our approach as it provides a computational algorithm to numerically approximate the solutions of ODEs. It allows us to solve ODE-based models when analytical solutions are not readily available or practical to compute. Commonly used numerical methods such as the Runge-Kutta method (Butcher, 1996) or the Euler method (Biswas et al., 2013) can be employed as ODE solvers in this context.

## 2.4 ELEMENT-WISED FILTERING

To enhance and refine the outcomes, we introduce an element-wise filter as a proposed approach. The element-wise filter is applied to the input $x(t_1)$ using a learnable filter matrix $K$. The operation is defined as follows:

$$\hat{x}(t_1) = K \odot x(t_1), \tag{11}$$

where $\odot$ represents the element-wise multiplication, also known as the Hadamard product. The element-wise filter matrix $K$ has the same dimensions as $x(t_1)$ and acts as a filter for individual elements. The resulting vector $\hat{x}(t_1)$ represents the refined output. For exploration purposes without introducing biases, we initialize the filter matrix $K$ with a uniform distribution, enabling exploration of the solution space. We show the pseudocode of our method in Algorithm 1.

---

**Algorithm 1** Pseudocode of FODE

---

**Input**: $t_0, t_1$, input data: $\{x_n\} = x_0, x_1, ..., x_n$
**Parameters**: $W, \theta$
**Construct** $f$: $x[n] \rightarrow z[n]$
    (FFT) $X[k] = \sum_{n=0}^{N-1} x[n] e^{-i2\pi kn/N}$
    $X[k]^{real}, X[k]^{imag} = real(X[k]), imag(X[k])$
    $X^{info} = X[k]^{real} \oplus X[k]^{imag}$
    $Z^{info} = g(X^{info}, \theta_g)$
    $Z[k]^{real} \oplus Z[k]^{imag} = Z^{info}$
    $Z[k] = Z[k]^{real} + iZ[k]^{imag}$
    (IFFT) $z[n] = \frac{1}{N} \sum_{k=0}^{N-1} Z[k] e^{-i2\pi kn/N}$
$x(t_1) = ODESolver(x(t_0), f, t_0, t_1, \theta)$
**output**: $\hat{x}(t_1) = K \odot x(t_1)$

---

## 3 Experiment

In this section, we conduct a comprehensive performance evaluation of the proposed Fourier Ordinary Differential Equations (FODE) model, comparing it with existing continuous models, namely NODE (Chen et al., 2018), ANODE (Dupont et al., 2019), SONODE (Norcliffe et al., 2020), and NCDE (Kidger et al., 2020), as well as discrete models including FNO (Li et al., 2020), RNN (Rumelhart et al., 1985) and LSTM (Hochreiter & Schmidhuber, 1997), on both time series forecasting and time series classification tasks.

For the time series forecasting tasks, we select three time series datasets from physics systems. Accurate forecasting of physics formulas is essential for understanding system behavior and making informed decisions. The chosen datasets cover Damped Oscillation, Forced Vibration, and Newton's Equations of Motion. Each model is trained for 50 epochs on each dataset for time series forecasting tasks. We sample 1000 points at equal intervals from 0 to 10 to create the full dataset. We split the dataset (Section 3.2) into a training set and a test set with a ratio of 0.75. For the time series forecasting task, we use a sliding window of length 50 to predict 50 future values based on 50 historical data.

Regarding the time series classification task, our focus is on Electrocardiogram (ECG) classification. ECG classification plays a crucial role in diagnosing and monitoring various cardiac conditions (Houssein et al., 2017; Pyakillya et al., 2017). By analyzing the electrical activity of the heart captured in ECG signals, healthcare professionals can identify abnormalities and determine appropriate treatments. We employ three real ECG datasets, namely ECGFiveDays, ECG200, and ECG5000, which are obtained from (Bagnall et al., 2018). For ECGFiveDays, we train each model for 200 epochs, while for ECG200 and ECG5000, we train each model for 100 epochs. Additional details regarding the datasets are provided in Section 3.2.

### 3.1 Environment Setup

For all experiments, we utilize Adam (Kingma & Ba, 2014) as the optimizer with a learning rate of $10^{-3}$ and a batch size of 32. We use the ReLU as the activate function. For all the ODE-based models, the solver method we used is "dopri5". For the time series forecasting tasks, we trained each model 50 epochs and used the MSE as the loss function. For the time series classification tasks, we trained each model 100 epochs and used the Cross-Entropy Loss as the loss function. For a fair comparison, we conduct the FNO with one Fourier layer where $modes = 2$ and $width = 8$. As seen in Table 2, the number of parameters in FNO is more than in FODEs. To ensure reliable results, we ran each experiment three times to account for experimental variability. The vector field in all the ODE-based models is parameterized using a 3-layer MLP (Haykin, 1998). These three layers have the dimension of $(F, H)$, $(H, H)$, and $(H, F)$, respectively, where the $F$ represents the number of features and $H$ represents the hidden dimensions set as $H = 16$. Our implementation is based on Python 3.8 and realized in PyTorch. The experiments were performed on an Apple M2 device equipped with an 8-core CPU.

### 3.2 DATASETS AND PROBLEM SETUP

**Damped Oscillation.** The damped oscillation (Smith, 1957) represents a system undergoing oscillatory motion with damping. The function describing the damped oscillation behavior is given by $x(t) = Ae^{-\beta t}(cos(\omega_1 t+\phi_1)+sin(\omega_2 t+\phi_2))$. Here, $x(t)$ represents the displacement of the system at time $t$, $A$ is the initial amplitude, $\beta$ is the damping coefficient, $\omega$ is the angular frequency, and $\phi$ is the phase angle. In our experiment, we set the function as $x(t) = e^{-t}(cos(\pi t+0.1)+sin(\pi t+0.1))$.

**Forced Vibration.** The forced vibration (Trifunac, 1972) represents a system vibrating under the influence of an external force. In this scenario, we consider a function expressed as the product of two sine functions: $x(t) = sin(\omega_1 t)sin(\omega_2 t)$. The values $\omega_1 = 0.5$ and $\omega_2 = 3$ are selected to define the angular frequencies associated with the sine components.

**Newton's Equations of Motion.** Newton's equations of motion, a fundamental principle in classical mechanics established by (Papapetrou, 1951), describe the relationship between the motion of an object and the forces acting upon it. In the context of one-dimensional motion along the x-axis, the position of the object as a function of time is given by $x(t) = x_0 + v_0 t + \frac{1}{2}at^2$. Simplifying this equation for our experiment, we employ the expression $x(t) = 0.1t + 0.5t^2$ to analyze the motion of the object.

**ECGFiveDays.** The ECGFiveDays dataset (Hu et al., 2013) consists of ECG recordings from a 67 year-old male. The dataset includes two distinct classes, corresponding to two different dates when the ECG measurements were recorded, with a five-day interval between them. Specifically, the dataset consists of 23 samples in the training set and 861 samples in the test set. Each time series in the dataset has a length of 136.

**ECG200.** The ECG200 dataset (Olszewski, 2001) is a binary classification dataset that captures the electrical activity during a single heartbeat. The two classes represent a normal heartbeat and an occurrence of myocardial infarction. Each time series in ECG200 has a length of 96. Both the training and test sets consist of 100 samples.

**ECG5000.** The ECG5000 dataset is a 20-hour-long ECG dataset obtained from the BIDMC Congestive Heart Failure Database (CHFDB) (Goldberger et al., 2000). The data were pre-processed by extracting individual heartbeats and equalizing their lengths using interpolation. The dataset consists of $5,000$ randomly selected heartbeats from a patient with severe congestive heart failure. There are 5 classes, and each time series has a length of 140. The training size is 500, and the test size is 4500.

### 3.3 TIME SERIES FORECASTING

In this section, we evaluate the performance of FODE compared to baseline models, including NODE, ANODE, SONODE, RNN, and LSTM for time series forecasting. The objective is to dynamically forecast the future behavior of a system based on historical data. We employ a sliding window approach with a length of $50$, aiming to predict the future $50$ values given the past $50$ values.

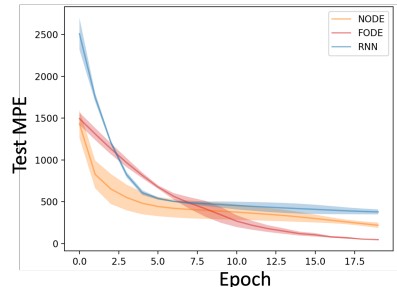

Figure 2: The test Mean Percentage Error (MPE) for damped oscillation system.

Figure 3 provides an intuitive comparison of FODE with the continuous model NODE and the discrete model RNN on three different systems: damped oscillation, forced vibration, and Newton's equations. FODE exhibits superior predictive accuracy across all systems. Notably, RNN fails to capture the dynamics of Newton's equations in the test set, while NODE achieves a reasonable fit but with less accuracy. Figure 2 presents the test Mean Percentage Error (MPE) during the training process for the damped oscillation system. It demonstrates that FODE consistently outperforms the other models as the training epoch increases.

We further compare our model with additional baseline models and record the test Mean Square Error (MSE) over three runs, as summarized in Table 1. Our FODE consistently achieves the best results for two out of three physical systems. Particularly, in the case of damped oscillation, FODE exhibits an error magnitude lower than the other models by at least one order of magnitude. The

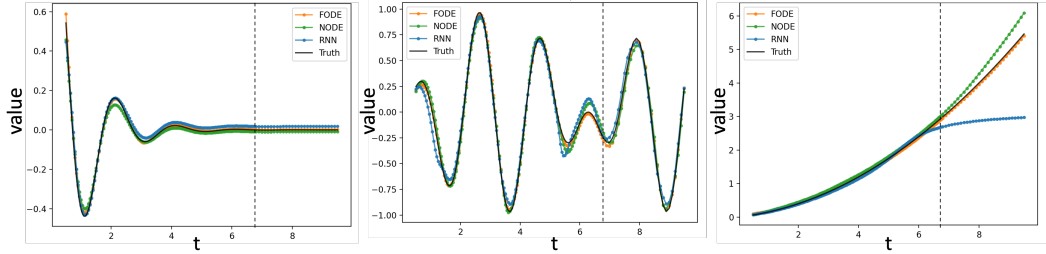

Figure 3: Performance comparison of FODE, NODE, and RNN on damped oscillation, forced vibration, and Newton's equations systems. The dotted line separates the training set from the test set.

Table 1: Testing MSE for Time Series Forecasting in Three Functions

|  | Damped oscillation $(\times 10^{-7})$ | Forced vibration $(\times 10^{-4})$ | Newton's Equations |
|---|---|---|---|
| RNN | $2105.21 \pm 38.70$ | $1506.26 \pm 484.95$ | $795.12 \pm 3.60$ |
| LSTM | $1877.76 \pm 130.40$ | $702.09 \pm 280.19$ | $766.79 \pm 5.97$ |
| NODE | $122.79 \pm 66.79$ | $267.50 \pm 3.73$ | $1.03 \pm 0.55$ |
| ANODE | $22.58 \pm 7.596$ | $275.35 \pm 8.57$ | $\mathbf{0.49 \pm 0.06}$ |
| SONODE | $36.51 \pm 37.95$ | $266.76 \pm 15.55$ | $1.23 \pm 0.23$ |
| **FODE** | $\mathbf{7.68 \pm 6.12}$ | $254.31 \pm 28.59$ | $0.68 \pm 0.09$ |
| **FODE (w/o K)** | $8.62 \pm 2.53$ | $\mathbf{230.96 \pm 19.51}$ | $1.80 \pm 0.48$ |

FODE model without filter K is denoted as FODE (w/o K). Table 1 shows that FODE performs better than FODE (w/o K) on 2 of the 3 datasets. FODE outperforms all models on all datasets except for ANODE on Newton's equations. However, it is noteworthy that FODE has fewer parameters compared to ANODE, with 506 parameters versus 556. A comparison of the number of parameters across different datasets is presented in Table 2. In the ODE-based models, ANODE and SONODE always have more parameters than ours.

## 3.4 TIME SERIES CLASSIFICATION

In this section, we apply FODE to time series classification tasks and compare its performance against baseline models including models in Section 3.3, NCDE (Kidger et al., 2020), and FNO (Li et al., 2020). We measure the test losses for the three datasets. Table 3 presents the MSE values for FODE and baseline models. It can be observed that FODE achieves comparable or lower MSE compared to the baselines.

Importantly, FODE exhibits a notable advantage in terms of parameter efficiency. Table 2 provides a comparison of the number of parameters, revealing that FODE has fewer parameters than SON-ODE and ANODE. For instance, SONODE has nearly twice the number of parameters compared to FODE. This advantage in parameter efficiency is a significant characteristic of ODE-based models (Chen et al., 2018). Excessive parameterization compromises this advantage.

**Hidden State Analysis of FODEs.** In order to gain insights into the iterative process of FODE, we conduct a short-time Fourier transform (Griffin & Lim, 1984) on the hidden state during model

Table 2: The number of parameters for each model for six datasets

|  | RNN | LSTM | NODE | ANODE | SONODE | NCDE | FNO | FODE | FODE (w/o K) |
|---|---|---|---|---|---|---|---|---|---|
| Three Physical Systems | 338 | 1250 | 423 | 556 | 860 | - | - | 506 | 456 |
| ECGFiveDays | 338 | 1250 | 595 | 900 | 1204 | 4306 | 2394 | 764 | 628 |
| ECG200 | 338 | 1250 | 515 | 740 | 1044 | 4306 | 1754 | 644 | 548 |
| ECG5000 | 389 | 1301 | 1026 | 1759 | 2063 | 4357 | 5821 | 1199 | 1059 |

Table 3: Test Loss for Time Series Classification

|  | ECGFiveDays | ECG200 | ECG5000 |
|---|---|---|---|
| RNN | $0.538 \pm 0.001$ | $0.583 \pm 0.002$ | $0.383 \pm 0.015$ |
| LSTM | $0.523 \pm 0.044$ | $0.605 \pm 0.010$ | $0.318 \pm 0.018$ |
| NODE | $0.192 \pm 0.034$ | $0.377 \pm 0.005$ | $0.254 \pm 0.010$ |
| ANODE | $0.189 \pm 0.027$ | $0.393 \pm 0.021$ | $0.253 \pm 0.008$ |
| SONODE | $0.178 \pm 0.040$ | $0.352 \pm 0.003$ | $\mathbf{0.247} \pm 0.001$ |
| NCDE | $0.714 \pm 0.022$ | $0.623 \pm 0.013$ | $0.913 \pm 0.004$ |
| FNO | $0.170 \pm 0.101$ | $0.330 \pm 0.011$ | $0.269 \pm 0.020$ |
| **FODE** | $\mathbf{0.114} \pm 0.072$ | $0.320 \pm 0.015$ | $0.556 \pm 0.426$ |
| **FODE (w/o K)** | $0.385 \pm 0.220$ | $\mathbf{0.317} \pm 0.026$ | $0.259 \pm 0.005$ |

training. The analysis reveals that FODE effectively maps the time series from one frequency spectrum to another. This finding indicates that FODE learns a mapping relationship in the frequency domain.

Figure 4 presents the results of applying the short-time Fourier transform to the hidden state of FODEs. The leftmost figure displays the frequency domain information of the original signal. As the model undergoes training, the frequency domain information is mapped to a different frequency domain space. FODE captures the dynamic mapping relationship in the frequency domain. It is worth noting that FODE leverages the Fourier transform in its implementation, and the short-time Fourier transform is utilized here to enhance the visualization of the signal's frequency domain information.

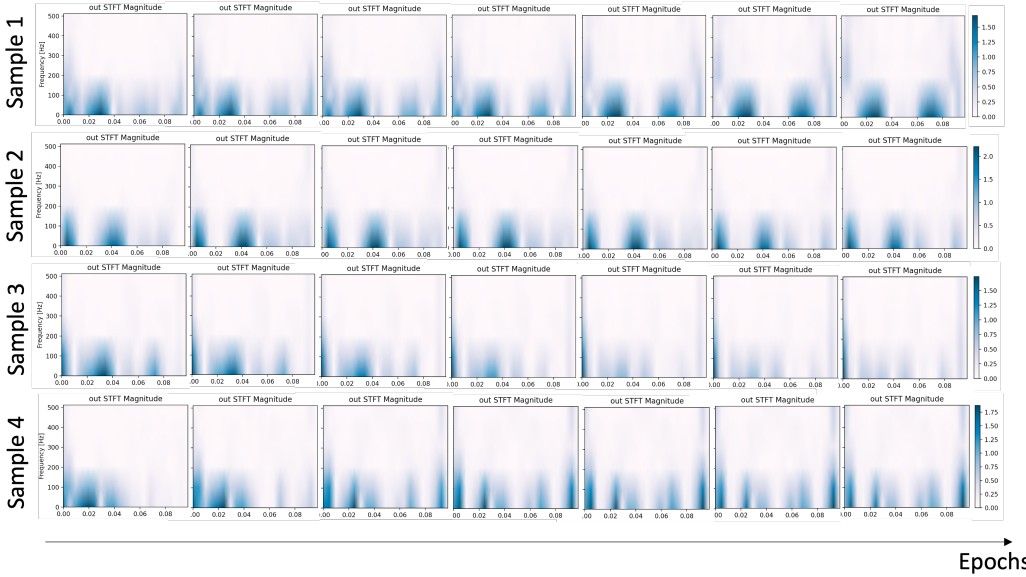

Figure 4: We apply the short-time Fourier transform on the hidden state of FODEs. The results show that FODE transforms a signal from one frequency domain to another.

**Evolution of filter K with different initialization** We present the evolution of element-wise filter values throughout the training epochs. We experiment with different initialization schemes for the element-wise filter values, namely all zeros, all ones, and Xavier uniform (Glorot & Bengio, 2010) initialization. The varying values of filter K are visually represented through color changes in Figure 5. On the left side, the corresponding loss is displayed in relation to the epoch.

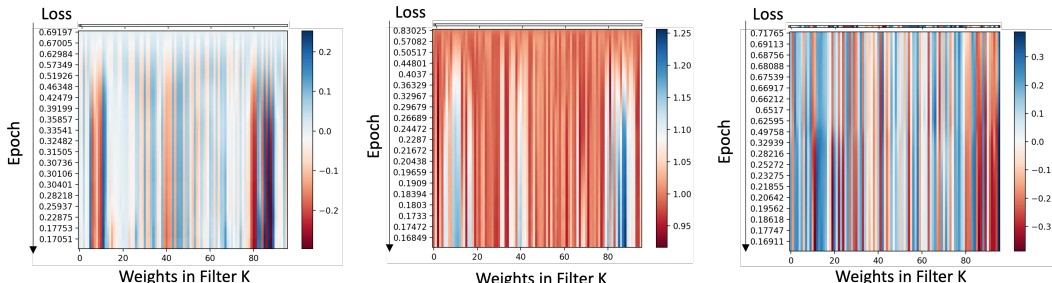

Figure 5: The evolution of filter K with different initialization. From left to right, K is initialized by Zeros, Ones, and Xavier Uniform, respectively. From top to bottom, each subfigure shows the timeline of the training process (the training loss is from high to low). The changing of colors represents the changing of weight values in K during training.

## 4 RELATED WORK

**Fourier Transform Differentiation.** The Fourier transform has been widely employed in spectral methods for solving differential equations, leveraging the equivalence between differentiation and multiplication in the Fourier domain. Its significance extends to the domain of deep learning, where it has been instrumental in the proof of the universal approximation theorem (Hornik et al., 1989) and demonstrated practical benefits, such as accelerating convolutional neural networks (Mathieu et al., 2013). These advancements underscore the profound impact of Fourier transforms in both theoretical and practical aspects of deep learning (Li et al., 2020; Holt et al., 2022). When applied to ordinary differential equations (ODEs), the Fourier transform enables the conversion of the equation into an algebraic equation in the frequency domain, which can be solved using standard techniques. Specifically, the Fourier transform of a function's derivative is proportional to the product of the Fourier transform of the function and the frequency variable. This property facilitates the transformation of a first-order ODE into an algebraic equation in the frequency domain. Consider a first-order ODE: $y'(t) = f(t)$. Taking the Fourier transform of both sides yields $i\omega Y(\omega) = F(\omega)$, where $Y(\omega)$ and $F(\omega)$ are the Fourier transforms of $y(t)$ and $f(t)$, respectively, and $\omega$ is the frequency variable. Rearranging this equation gives $Y(\omega) = \frac{F(\omega)}{i\omega}$, which can be inverted back to the time domain to obtain the solution $y(t)$.

**Hadamard Product.** The Hadamard product-based layer has garnered attention in various domains of machine learning and data analysis due to its adaptability and compatibility (Gama et al., 2018; Ngiam et al., 2011; Trask et al., 2018). For instance, Xing et al. (2018) propose a convolutional neural network with element-wise filters (CNN-EW) for brain networks, achieving improved accuracy in distinguishing subject groups and identifying abnormal brain regions associated with autism spectrum disorder (ASD). Rao et al. (2021) present the Global Filter Network (GFNet), an efficient architecture that learns long-term spatial dependencies in the frequency domain with log-linear complexity. Chen et al. (2019) propose a novel convolutional neural network with element-wise filters for classifying dynamic functional connectivity (DFC-CNN). Furthermore, Dzanic & Witherden (2022) utilize element-wise filtering to mitigate spurious oscillations near discontinuities in discontinuous spectral element methods.

## 5 CONCLUSION

In this work, we introduced FODE, a novel ODE-based model that leverages the Fourier domain to learn dynamics and enhance the representation of time series data. By operating in the Fourier space, FODE effectively captures underlying patterns and features, surpassing the capabilities of traditional continuous models. The incorporation of an element-wise filter maintains granularity while enabling generalization. Experimental evaluations on various time series datasets demonstrated the superior performance of FODE, achieved with a reduced parameter footprint. In future research, we plan to extend FODE to other domains and explore its interpretability and robustness properties.

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
