# OpenReview forum: "Fourier Ordinary Differential Equations"
_ICLR.cc/2024/Conference — Submitted to ICLR 2024_

### Official Review · Reviewer_cX5F · 2023-10-25

**Soundness:** 3 good
**Presentation:** 3 good
**Contribution:** 3 good
**Rating:** 6
**Confidence:** 4

**Summary:**

The paper looks to learn a Neural ODE (NODE) system that resides in the Fourier domain, and shows promising empirical results on some time series analysis tasks. The main contribution is to learn NODEs in the Fourier domain rather than Eucledian domain.Also, with element-wise filtering, FODEs address biases in continuous models.

**Strengths:**

The empirical results are promising, and the presentation is clear. The novelty mainly comes from learning NODE systems in the Fourier domain, and is well investigated.

**Weaknesses:**

Despite the obvious performance boost compared to NODEs and RNNs, there are many SOTA benchmarks not addressed. For example, [1] has SOTA performances on many time series tasks, along with many other NODE methods. The reviewer understands that the point of the paper is to show learning in the Fourier domain is beneficial, but it would make the paper stronger if such comparisons are made. Also, compared to vanilla NODEs, what theoretical benefits do FODEs have? It would be great to understand this too.


[1] Kidger, Patrick et al. “Neural Controlled Differential Equations for Irregular Time Series.” ArXiv abs/2005.08926 (2020): n. pag.

**Questions:**

N/A

---

> ### Author Response · Authors · 2023-11-22
> **Response to reviewer**
>
> We appreciate the reviewer's recognition of our work. We also thank you for the feedback regarding the experimental validation of our study.
>
> We have conducted additional comparative experiments to support the superiority of the FODE model. We compare FODE with NCDE[1] and FNO[2] on time series classification tasks.  For a fair comparison, FNO is conducted with one Fourier layer where modes=2 and width=8. Despite this configuration resulting in the FNO having more parameters than our FODE model, as detailed in Table 2, our FODE model demonstrates superior performance. The results, presented in Table 3, show that our model achieves the lowest test loss on two out of the three ECG datasets used in our study.
>
> [1] Kidger, P., Morrill, J., Foster, J. and Lyons, T., 2020. Neural controlled differential equations for irregular time series. Advances in Neural Information Processing Systems, 33, pp.6696-6707.
>
> [2] Li, Zongyi, et al. "Fourier Neural Operator for Parametric Partial Differential Equations." International Conference on Learning Representations. 2020

---

> > ### Comment · Reviewer_cX5F · 2023-11-23
> >
> > Thanks for the reply, I will keep my score as is.

---

### Official Review · Reviewer_MSqR · 2023-11-01

**Soundness:** 3 good
**Presentation:** 3 good
**Contribution:** 2 fair
**Rating:** 5
**Confidence:** 4

**Summary:**

The paper proses a model dynamics in the frequency domain for an ODE. This builds along the line of NODE and many other works. The two main components as mentioned in the paper are the use of Fourier transform and hence the modelling of the dynamics in the frequency space and finally the use of an element-wise filter prior to predicting the output. Empirical evaluation is provided on several time-series modelling tasks as an evidence for the model performance.

**Strengths:**

The paper studies the utility of modelling dynamics in the frequency space, specifically using Fourier transform for ODEs while enabling learning via neural networks.

It is well articulated, making it easy to understand and follow through.

The element-wise learnable filter seems to be improving performance and I believe can be a good add-on to other models belonging to the same family.

**Weaknesses:**

The use of Fourier transform and modelling the dynamics in the frequency space have been demonstrated for PDEs [1] and has been widely used, given that the extension to ODE and hence performing reasonably well enough is expected. What I don't find is authors clearly pointing out the value in their work and operator learning framework and/or even the frequency space modelling used for ODE in [2]. Note that [2] uses Laplace transform, but I think this in principle achieving the same modelling paradigm as FODE. Comparing and contrasting to these is critical I think.

Furthermore, the experimental results are okay, but not very convincing that the. proposed method has significant benefits, if any.

Please refer to the questions, below for more specific.

[1] Li, Zongyi, et al. "Fourier Neural Operator for Parametric Partial Differential Equations." International Conference on Learning Representations. 2020

[2] Holt, Samuel I., Zhaozhi Qian, and Mihaela van der Schaar. "Neural Laplace: Learning diverse classes of differential equations in the Laplace domain." International Conference on Machine Learning. PMLR, 2022

**Questions:**

1) I am not sure why the authors mention "Euclidean space" as a contrast to their frequency space modelling method. I don't think this is correct, specifically is FODE using any metric space different from "Euclidean", shouldn't it be just contrasted by mentioning in the data space or something like that? If one uses to choose specifics, these need to be pointed out, mentioning terms vaguely isn't helpful

2) It is mentioned that "time differentiation in the Fourier domain simplifies the modeling of dynamics as it becomes a multiplication operation". I believe author's refer to the convolution property in this case. Is this even being used implicitly and/or explicitly if not, what is the point of mentioning some property. DFT has many properties, if they are not used, these shouldn't be presented as if they are helping the proposed method in any way.

3) What is $T$ in equation (1)? Should it be $t_1$ instead?

4) Eqn (3) is wrong, I will consider this as typo.

5) So, DFT from eqn (4) is being performed along the feature dimension, if this is the case, what do frequency even mean in this case? If authors can clarify this, it would be great.

6) In experimental result, comparison to a frequency space based method is something that I would like to see. If nothing else [2] can be a good baseline, any comment on this one?

7) I would expect time-series classification results be evaluated on accuracy and not MSE, table 3. Why is the choice for MSE justified for classification task?

8) In Table (2) what is the result for NODE with comparable parameters, can the authors discuss this briefly if possible?

9) Figure 4 is good, this is expected, but is there any further analysis that authors plan to present?

10) I am not sure about the utility of Figure 5, this needs to be explained and justified better.

---

> ### Author Response · Authors · 2023-11-22
> **Response to questions**
>
> We appreciate the questions the reviewers raised. We'd like to provide further clarification and insight into these questions.
>
> Q1: Euclidean space in our paper is just used to contrast to the frequency space. We realized it may cause confusion, so we changed it to “Time domain” in the revision.
>
> Q2: We appreciate the review’s question but we do not refer to convolution property here. We aim to say When one transforms a signal into the Fourier domain (using the Fourier Transform), differentiation with respect to time becomes a multiplication by the frequency variable. Mathematically, if f(t) is a time-domain function and F(ω) is its Fourier transform, then the Fourier transform of the derivative df(t)/dt​ is given by jωF(ω), where j is the imaginary unit and ω represents frequency. It is actually an important motivation for our work. Based on these properties, we plan to learn the derivative df(t)/dt​ in the Fourier domain and then use it for ODE based model.
>
> Q3 & Q4:
> Thanks for pointing them out. Yes, these are some typos, and we corrected them in the revision.
>
> Q5:
> In our setting, the DFT(FFT) from equ (4) is being performed along the time series length dimension. Thus, I think the frequency of it will be easily understandable.
>
> Q6:
> We realize that compared with frequency space-based method is important to support the superiority of the FODE model.
> we have conducted additional comparative experiments to support the superiority of the FODE model. We further compare FODE with FNO[1] (belongs to the frequency space-based method) and Neural Controlled ODE (Other reviewers mentioned) on time series classification tasks.  We read [2] and we think it is a good baseline, we added [2] in related work since the time limit.
> For a fair comparison, FNO[1] is conducted with one Fourier layer where modes=2 and width=8.
> Despite this configuration resulting in the FNO having more parameters than our FODE model, as detailed in Table 2, our FODE model demonstrates superior performance. The results, presented in Table 3, show that our model achieves the lowest test loss on two out of the three ECG datasets used in our study.
>
> Q7:
> This is a typo. For time-series classification, we use Cross-Entropy Loss as the loss function, and the test loss is what we recorded during the experiment. Thus, table 3 describes the “Cross-Entropy Loss ”. Please see the revision.
>
> Q8:
> The NODE with comparable parameters will still have worse performance than FODEs. We can be convinced of this point from our rich empirical experiments. Also, even ANODEs and SONODEs are more advanced models and always have better performance than original NODEs with comparable parameters. We also compared more models as shown in Table 3.
>
> Q9:
> From Figure 4, we can see how hidden states evolve in the frequency domain for time series. The short-time Fourier transform is utilized here to enhance the visualization of the signal’s frequency domain Information.
>
> Q10:
> Figure 5 is used to illustrate how the filter K be trained during training. The evolution of filter K with different initialization. From left to right, K is initialized by Zeros, Ones, and Xavier Uniform,  respectively. From top to bottom, each subfigure shows the timeline of the training process (the training loss is from high to low). The changing of colors represents the changing of weight values in K during training. We added more descriptions in the revision.
>
> [1] Li, Zongyi, et al. "Fourier Neural Operator for Parametric Partial Differential Equations." International Conference on Learning Representations. 2020
>
> [2] Holt, Samuel I., Zhaozhi Qian, and Mihaela van der Schaar. "Neural Laplace: Learning diverse classes of differential equations in the Laplace domain." International Conference on Machine Learning. PMLR, 2022

---

> > ### Comment · Reviewer_MSqR · 2023-11-22
> >
> > Thank you authors for clarifying some of the points. I still feel, the paper would significantly benefit from a comparison with [2] above, this is the closest and most reasonable baseline to show the advantage of FODE. Furthermore, I still don't see a revision in the text for table 3, there is still mention of MSE. Also, it is much easier to interpret, if authors report accuracy for classification tasks. I am not sure, why authors stick to reporting cross-entropy loss, despite previous comment. I don't think this would take anymore time than just loading a trained model and running inference. At this point, I am reluctant to increase my score.

---

### Official Review · Reviewer_GqMG · 2023-11-05

**Soundness:** 2 fair
**Presentation:** 2 fair
**Contribution:** 2 fair
**Rating:** 3
**Confidence:** 4

**Summary:**

In this work, authors present a Fourier-based framework to learn the dynamics of systems following ODEs. Essentially, time-series systems are learned by first applying an FFT followed by an inverse FFT. Then an element-wise product with a filter is applied to obtain the updated field variable. The resulting framework is termed as FODE. The empirical studies on six different datasets including three datasets on ECG show superior results over the baselines.

**Strengths:**

The following strengths of the work.

S1. Learning dynamical time-series system is an important problem and the work makes a contribution towards this direction.

S2. The idea of combining Fourier transform with an ODE is interesting.

S3. Empirical results show superior performance of FODE over the chosen baselines.

**Weaknesses:**

There are several weaknesses for the work as follows.

W1. The idea of using Fourier transform for learning dynamical systems is not new. There are a large family works following Fourier Neural Operators, Deep Operator Networks, and Koopman operators in this direction. Authors have not discussed or compared the results with any of these works in the manuscript.

W2. The experimental systems taken in the work are not very complex. In NODE-based works and in FNOs, much more complex time-series systems such as fluid flow, weather forecasting, complex dynamical equations etc. are considered. However, the first three examples considered in this work are fairly simple equations resulting in 1D dynamics. The ECG dataset is a realistic dataset that is reasonably complex. However, this is again 1D. It is not clear how well the system will scale to larger and more complex problems.

W3. The baselines considered are fairly simple ones that are not truly SOTA. Authors should compare the results with FNO, DeepONet, or such frameworks which are considered to be SOTA.

W4. NODE is essentially a physics-informed approach which requires the integration of the equation of motion to obtain the time-series modeling. FNO is a purely data-driven approach which allows to predict $x_{t+1}$ directly from $x_{t}$ without any integration. By combining Fourier approach with ODE, it seems that the disadvantages of the NODE are still retained. That is, the time integration is still limited by the time-step and hence rollout over a large time window might be challenging.

W5. There is no discussion on the limitations of the present work. This should be included.

**Questions:**

Q1. What is the loss function? This should be clearly articulated.

Q2. More baselines should be compared. How well does the model perform in comparison to FNO, DeepONet and other SOTA time series models. RNNs and LSTMs are some of the earliest models to be considered as baselines.

Q3. How well does the model scale to more complex data? It is not clear.

Q4. What are the advantages of the FODE over NODE? Is it data-efficient? How robust is it to noisy data?

Q5. Although FODE seems to have a reduced parameter footprint over NODE, what about the training and inference time?

Q6. Why is a sliding window approach required? Can't the model predict the $x_{t+1}$ directly from $x_t$? Please clarify? If this is a hyperparameter, how does the model work when the only data point is used to predict the future? In the original NODE and FNO, only one data point is used for the future prediction.

**Details Of Ethics Concerns:**

Real world datasets on ECG are used. This seems to be used from previous research. In any case, the relevant ethical approval may be needed to use this data.

---

> ### Author Response · Authors · 2023-11-22
> **Response to questions**
>
> We appreciate the questions the reviewers raised. We'd like to provide further clarification and insight into these questions.
>
> Q1:
> For the time series forecasting tasks, MSE is used as the loss function. For the time series classification tasks, the Cross-Entropy Loss is used as the loss function.  We recognize the importance of providing comprehensive details about our experimental procedures. We provide more experimental setup details in the revision, please see the environment setup section.
>
> Q2:
> we have conducted additional comparative experiments to support the superiority of the FODE model. We compare FODE with NCDE[1] and FNO[2] on time series classification tasks.
> For a fair comparison, FNO is conducted with one Fourier layer where $modes=2$ and $width=8$.
> Despite this configuration resulting in the FNO having more parameters than our FODE model, as detailed in Table 2, our FODE model demonstrates superior performance. The results, presented in Table 3, show that our model achieves the lowest test loss on two out of the three ECG datasets used in our study.
>
> Q3:
> We agree with the reviewer that how well the model scales to more complex data is an important question.
> In this work, we plan to demonstrate a new method that learns the derivative df(t)/dt​ in the frequency domain and we test two tasks on several datasets to show this idea can work. We will test more complex datasets in our future work.
>
> Q4:
> There are several advantages of the FODE over NODE. First is the accuracy, as shown in Table 1 and Table 3, the performance of FODE is much better than NODE.  Second, the FODE provides an innovative way to calculate the derivative df(t)/dt​ in the frequency domain which may be an efficient way to capture the long-term dependence of time series data, and our experiments verify it.
>
> Q5:
> We did not record the training and inference time of FODE and other models. From our experience, basically, the training time of FODE will be a little more than NODE, depending on the specific datasets.
>
> Q6:
> In our paper, there are two experiment tasks. One of them is time series forecasting, as the reviewer mentioned.  In the time series forecasting tasks, we try to predict future values based on these historical values. The number of future values and historical values can be more than 1, for example, we use 50 historical values to forecast values in the future. The reason we use a sliding window approach is to provide a more flexible way to divide the data for this task. When the only data point is used, the FFT transform can be applied to the feature dimension. However, if the feature dimension is 1, our model currently cannot handle this situation. But we left the forecasting task based on just one initial point as a future work.
>
> [1] Kidger, P., Morrill, J., Foster, J. and Lyons, T., 2020. Neural controlled differential equations for irregular time series. Advances in Neural Information Processing Systems, 33, pp.6696-6707.
>
> [2] Li, Zongyi, et al. "Fourier Neural Operator for Parametric Partial Differential Equations." International Conference on Learning Representations. 2020

---

> > ### Comment · Reviewer_GqMG · 2023-12-02
> > **Thank you**
> >
> > Thank you for the responses. I will maintain my score.

---

### Official Review · Reviewer_hxik · 2023-11-09

**Soundness:** 2 fair
**Presentation:** 2 fair
**Contribution:** 2 fair
**Rating:** 5
**Confidence:** 4

**Summary:**

This paper introduces Fourier ordinary differential equations (FODEs), a new model that incorporates Fourier transforms and ordinary differential equations for analyzing time series data.The advantage of the FODEs model is its ability to capture the underlying patterns in the data through frequency component analysis, which in turn provides a better representation of the data and an in-depth understanding of the intrinsic dynamics of the system. The model reduces the difficulty of modeling complex systems by simplifying the differentiation operation to a multiplication operation in the Fourier domain, making it simpler to capture short-term and long-term dependencies in the data. An element-by-element filtering technique is also introduced in the article to keep the data fine-grained and to exploit the generalization capabilities of continuous models, an innovation that is crucial for the accuracy of time series analysis. Through experimental comparisons, FODEs outperform existing continuous and discrete time series models in terms of accuracy and generalization ability, and are more economical in terms of the number of parameters, demonstrating efficiency in terms of computational resources and training data requirements. These features make FODEs a promising tool in the field of analysis and prediction of complex dynamic systems.

**Strengths:**

The strength of the Fourier Ordinary Differential Equations (FODEs) model lies in its innovative integration of the Fourier transform with ordinary differential equations to analyze time series data. This hybrid approach allows for capturing the underlying patterns in data by analyzing frequency components, offering potentially better data representation and insight into the inherent dynamics of the systems being modeled. By converting differentiation into a multiplication operation in the Fourier domain, FODEs simplify the complexity of modeling dynamic systems, making it easier to capture both short-term and long-term dependencies in the data.

**Weaknesses:**

1. Problems of model generalization and applicability: Although the article proposes a new method for learning dynamics in the Fourier domain, the generalization and applicability of the method on different types and sizes of time series data are not fully discussed. In particular, the performance and applicability of the FODE model on time series data that are non-periodic or have relatively high signal noise is still a question mark. In addition, the Fourier transform is usually weak for non-smooth signals, which may limit the ability of FODE on complex or non-linear time series data.

2. Model Complexity and Computational Efficiency Issues: Despite the paper's claim that FODE improves on computational efficiency, in reality, the computational complexity of the Fourier transform and inverse transform may increase dramatically as the amount of data increases, especially on real-time or large-scale datasets. The paper lacks an assessment of the computational efficiency of a wider range of computations, such as on multidimensional datasets and the specific advantages and disadvantages compared to other existing methods.

3. Insufficient experimental validation and comparative analysis: while the paper provides some experimental results to support the superiority of the FODE model, it lacks comparative experiments, especially with the latest time series analysis methods. The article does not provide sufficient details of the experimental setup, such as the choice of hyperparameters for the model, training details, and measures to prevent overfitting. In addition, the article does not discuss the confidence intervals or statistical significance of the model in terms of forecasting performance, which reduces the persuasiveness of the experimental results.

**Questions:**

See Weaknesses

---

> ### Author Response · Authors · 2023-11-22
> **Resonse to the Weaknesses**
>
> 1. Thank you for raising these important concerns. We carefully considered these questions during the design of our model.
> We carefully considered these questions during the design of our model. We acknowledge that non-periodic and noisy signals can pose challenges to the FODE model's performance and applicability. To address these issues, we incorporated an element-wise filter into our model's architecture. The element-wise filter is a crucial component of our approach, designed to enhance the model's ability to handle various types of time series data, including those that are non-periodic or have high levels of signal noise. This filter allows the FODE model to adapt and capture the underlying dynamics even in challenging conditions.
>
> 2. We appreciate your concerns regarding the computational efficiency of our approach, and we'd like to provide further clarification and insight into our efforts to address these issues.
> (1) we use the FFT and IFFT inside the “ODE Function” and it makes sure the dimensions of input and output dimensions remain the same with a Neural ODE. For example, if the input data has a shape of (d1, d2) = (length, #features) after applying the FFT, the tensor’s shape becomes (d1’, d2’) = (length, #features//2), where d2’ contains real part and Imaginary part.
> (2) we use the Fast Fourier Transform (FFT) to reduce the computational complexity of DFT.
> (3) the integration of FFT and IFFT inside the 'ODE Function', coupled with their interaction with the ODE solver, forms a synergistic relationship. This synergy not only improves computational efficiency but also aids in effectively learning the vector field. As a result, it can potentially reduce the number of function evaluations required, further contributing to the model's overall efficiency.
>
> 3. Thank you for your feedback regarding the experimental validation of our study.
> (1) we recognize the importance of providing comprehensive details about our experimental procedures. we provide more experimental setup details in the revision, please see the environment setup section, including the number of epochs, loss function, ODE solver method, details of layers, etc. Algorithm 1 also shows the detailed implementation of our model.
> (2) we have conducted additional comparative experiments to support the superiority of the FODE model. We compare FODE with NCDE[1] and FNO[2] on time series classification tasks. For a fair comparison, FNO is conducted with one Fourier layer where modes=2 and width=8. Despite this configuration resulting in the FNO having more parameters than our FODE model, as detailed in Table 2, our FODE model demonstrates superior performance. The results, presented in Table 3, show that our model achieves the lowest test loss on two out of the three ECG datasets used in our study.
>
> [1] Kidger, P., Morrill, J., Foster, J. and Lyons, T., 2020. Neural controlled differential equations for irregular time series. Advances in Neural Information Processing Systems, 33, pp.6696-6707.
> [2] Li, Zongyi, et al. "Fourier Neural Operator for Parametric Partial Differential Equations." International Conference on Learning Representations. 2020

---

> > ### Comment · Reviewer_hxik · 2023-11-22
> >
> > Thank you for responding to my question, but not enough for me to give acceptable advice, I maintain my original score

---

### Meta-Review · Area_Chair_xzEf · 2023-12-07

**Metareview:**

The paper introduces a framework for learning ordinary differential equation (ODE) dynamics in the frequency domain to model time series. The objective is akin to that of Neural ODE, but the dynamics are represented in Fourier space. The model is constructed around two key components: encoding in Fourier space through Fast Fourier Transform (FFT) in the time dimension, modeling the dynamics, and returning to the original time space using inverse FFT. Subsequently, a pointwise filter is applied to compute the final output. The model's performance is assessed in time series forecasting and classification tasks.

The reviewers acknowledge the relevance and significance of the contribution. However, they raised several concerns regarding the paper's positioning in relation to existing literature on modeling dynamics in spectral spaces. Additionally, they emphasized the need for more experimental comparisons, particularly involving more complex datasets and new baselines. Despite its inherent interest, the paper is still to be strengthened by addressing these issues.

**Justification For Why Not Higher Score:**

Not ready for publication

**Justification For Why Not Lower Score:**

a

---

### Decision · Program_Chairs · 2024-01-16

Reject